# R-MADDPG for Partially Observable Environments and Limited Communication

**Rose E. Wang** [1]   **Michael Everett** [2]   **Jonathan P. How** [3]

## Abstract

There are several real-world tasks that would benefit from applying multiagent reinforcement learning (MARL) algorithms, including the coordination among self-driving cars. The real world has challenging conditions for multiagent learning systems, such as its partial observable and nonstationary nature. Moreover, if agents must share a limited resource (e.g. network bandwidth) they must all learn how to coordinate resource use. This paper introduces a deep recurrent multiagent actor-critic framework (R-MADDPG) for handling multiagent coordination under partial observable settings and limited communication. We investigate recurrency effects on performance and communication use of a team of agents. We demonstrate that the resulting framework learns time-dependencies for sharing missing observations, handling resource limitations, and developing different communication patterns among agents.

## 1. Introduction

To apply reinforcement learning in real world settings, we must develop robust frameworks that explicitly address common real world challenges. Much of current RL research makes unrealistic assumptions, like full observability of the environment, one agent learning in isolation, or unlimited access to a communication network, none of which exist in the real world. Therefore, RL algorithms must address the following three challenges inherent to real-world domains: partial observability (agents must learn concise abstractions of history while learning to make good decisions), nonstationarity (introduced by multiple agents learning simultaneously), and limited communication between agents (constraints on sharing of beliefs and intents). Example applications include search and rescue scenarios with constrained vehicle sensors (partial observability), cooperation between humans and machines (nonstationarity), space exploration missions or coordination among independent self-driving cars (limited communication).

Common solutions to address these challenges include multiagent learning (Foerster et al., 2017; Lowe et al., 2017), communication (Peng et al., 2017; de Freitas, 2016), and resource sharing. An extensive discussion on previous works will follow in Section 2.

This work proposes a new model, the recurrent multiagent deep deterministic policy gradient model (R-MADDPG), for handling multiagent coordination under partially observable environments using only limited communication, and compares the proposed architecture's performance against alternatives. R-MADDPG learns two policies in parallel– one for physical navigation and another for communication– and not individually as done in previous works. This work extends upon previous work, Multi-Agent Actor-Critic for Mixed Cooperative-Competitive Environments (MADDPG) (Lowe et al., 2017).

Specifically, we assume a multiagent actor-critic model and propose a model where both the actor and the critic are recurrent. Alternative architectures include actor-critic models with only a recurrent actor or only a recurrent critic. Our experiments show that the fully recurrent actor-critic model learns with less variability in mean and variance and that the recurrent critic is the crucial component that enables learning under real-world conditions (partial observations, limited communication, multiagent). The experiments suggest a recurrent actor is insufficient by itself for partially observable domains.

Our contributions include: i) a demonstration of the failure of current MARL methods in a simple partially observable coordination task, which identifies a remaining gap between RL research and the real world; ii) recurrent multiagent actor-critic architectures for message passing and movement, with experiments showing successful learning under various constraints on communication and observability; iii)

---

[1]Department of Electrical Engineering and Computer Science [2]Department of Mechanical Engineering [3]Laboratory for Information and Decision Systems; Massachusetts Institute of Technology, Cambridge, Massachusetts. Correspondence to: Rose E. Wang <rewang@mit.edu>.

*Reinforcement Learning for Real Life (RL4RealLife) Workshop in the 36th International Conference on Machine Learning*, Long Beach, California, USA, 2019. Copyright 2019 by the author(s).

empirical comparison between the proposed architectures that highlights the importance of a recurrent critic; and iv) an open-sourced implementation of R-MADDPG [1].

## 2. Related Works

Three key challenges in applying reinforcement learning to real life are: multiagent learning in fully and partially observable environments, multiagent learning for communication and/or communication protocols, and multiagent resource sharing. Most works below handle these challenges separately. This work is the first to handle all three of these challenges in one general framework.

**Multiagent learning** General multiagent reinforcement learning (MARL) methods either assume full observability and are less applicable to real world conditions (Peng et al., 2017; Kong et al., 2017), or handle partially observable environments by making assumptions on the types of policies learned, such as multiple agents developing homogeneous policies (Khan et al., 2018). Earlier works (Wu et al., 2009; Amato et al., 2015) model the multiagent learning problem as decentralized POMDPs (Dec-POMDPs), nonetheless the traditional search for an optimal policy requires knowledge about the transition function which agents typically do not have access to in the real world.

A well-known issue in multiagent learning is nonstationarity (Hernandez-Leal et al., 2017): Each agent simultaneously updates its policy during training, thus making each agent's optimal policy a moving target. MADDPG combats nonstationarity by training the critic in a centralized manner, as in this work[2]. Several single agent RL works address nonstationarity with experience replay (Mnih et al., 2015; Schaul et al., 2015). However, experience replay in multiagent setting introduces additional challenges, such as how to sample experiences in a synchronized fashion (Omidshafiei et al., 2017), and even conflicting information about whether experience replay helps in multiagent settings (Foerster et al., 2016; Singh et al., 2018).

**Communication and resource sharing** Previous multiagent communication methods miss important elements of the real world: the architectures are designed specifically for communication (de Freitas, 2016) and assume network parameter sharing (Foerster et al., 2016) or access to other agents' hidden states (Singh et al., 2018; Sukhbaatar et al., 2016). Not only are these assumptions unrealistic for real world conditions, enforcing a specific communication architecture can limit the diversity of emergent communication protocols (Kottur et al., 2017).

---

[1] https://github.com/rosewang2008/rmaddpg
[2] This work distinguishes between centralized training (sharing experiences during network parameter updates) and communication messages (sharing observations/beliefs during task execution).

Other works model communication separately from other task policies (like physical motion) (Khan et al., 2019) even though it oftentimes aids those objectives and should be learned in conjunction, or propose models for learning long-term sequential strategies (Peng et al., 2017) but condition on complete state information. Previously mentioned works have used recurrence in multiagent reinforcement learning and communication, but they do not use it for message passing or simultaneously modelling communication and other task policies.

MADDPG handles cooperative tasks, however does not model explicit communication among agents and cannot handle partially observable environments and history-dependent decision making. (Khan et al., 2018) is similar to MADDPG, however scales better to more agents under the strong assumption that the agents' policies can be approximated to a single policy. (Jiang & Lu, 2018) is similar to our learning environment, in that they want to learn how to conservatively use communication. They propose a central attentional unit in an actor-critic framework for learning when communication is needed and for integrating shared information. Nonetheless, they prioritize minimizing communication as much as possible, whereas this paper demonstrates that agents are capable of adapting to any amount of resources.

## 3. Background

### 3.1. Reinforcement Learning

In real world settings, agents make noisy observations of the true environment state to inform their action selection, typically modeled as a Partially Observable Markov decision process (POMDPs) (Kaelbling et al., 1998), or in its extended version with multiple agents, a Decentralized Partially Observable Markov decision process (Dec-POMDPs) (Bernstein et al., 2002) defined as $(\mathcal{I}, \mathcal{S}, \mathcal{A}, \mathcal{T}, \Omega, \mathcal{O}, \mathcal{R}, \gamma)$, where $\mathcal{I} = \{1, ..., N\}$ is the set of $N$ agents, $\mathcal{S}$ is the set of states, $\mathcal{A} = \times_i \mathcal{A}_i$ is the set of joint actions, $\mathcal{T}$ is the transition probability function, $\Omega = \times_i \mathcal{O}_i$ is the set of joint partial observations, $\mathcal{O}$ is the observation probability function, $\mathcal{R}$ is the reward function, and $\gamma \in [0, 1)$ is the discount factor. At each timestep $t$, agent $i$ receives a partial observation $o_t^i$ and takes action $a_t^i$ according to policy $\pi^i(h_t^i; \theta^i)$, where $\theta^i$ is agent $i$'s policy parameters and $h_t^i$ is agent $i$'s observation history. The current state of the Dec-POMDP $s_t$ transitions to $s_{t+1}$ according to the transition function with joint actions of the agents $a_t = a_t^1 \times ... \times a_t^N$, i.e. $\mathcal{T}(s_{t+1}; s_t, a_t)$. The agents receive a shared team reward $r_t = \mathcal{R}(s_t, a_t)$, and receive a new joint observation set $o_{t+1} = \{o_{t+1}^1, ..., o_{t+1}^N\}$ after the state transition. The objective for each agent is to maximize its expected discounted reward $\mathbb{E}[\sum_t r_t \gamma^t]$.

This work focuses on using recurrent neural networks for

learning representations capable of estimating the true state of the Dec-POMDP $\mathcal{S}$ from an agent's local set of observations $\Omega_i$. The recurrency in the network architecture therefore explicitly acts as a system mechanism for gathering partial observations so as to minimize the differences in system behavior with and without full observability of $\mathcal{S}$.

## 3.2. Q-learning

Q-learning and Deep Q-learning methods have been very popular in the context of Atari game playing. Q-learning is a model-free approach for determining the long-term expected return of executing an action $a$ from a state $s$, where it makes use of the action-value function under a given policy $\pi$ (Sutton et al., 1998). In other words, Q is iteratively defined as,

$$Q_\pi(s, a) = \mathbb{E}_{s'}[r(s, a) + \gamma \mathbb{E}_{a' \sim \pi}[Q_\pi(s, a)]]. \quad (1)$$

Deep Q-Learning methods approximate the Q-values by means of a neural network parameterized by the weight $\theta$. It learns the values for $Q^*$, where $\tilde{Q}^*$ is the target values, by minimizing the loss defined as:

$$\mathcal{L}(\theta) = \mathbb{E}_{s,a,r,s'}[(Q^*(s, a|\theta) - (r + \gamma \max_{a'} \tilde{Q}^*(s', a')))^2]. \quad (2)$$

Because the same network is used for generating next target values and for updating $Q^*$, Deep Q-Learning demonstrates high variance in its learning trajectory for approximating action values. Thus, common techniques for facilitating learning stability include using experience replay (Mnih et al., 2015; Schaul et al., 2015) in a replay memory buffer sampled during training, and using a separate, target network $\tilde{Q}$ for generating the target values in the loss calculation. This target network is identical to the $Q^*$ except that the target network is updated to match $Q^*$ at a much slower rate (e.g. every thousand iterations) so as to stabilize the learning of $Q^*$.

## 3.3. Policy Gradient Algorithms

Policy gradient methods are another way for maximizing expected reward for the agent by directly optimizing the policy. The policy is parameterized by weights $\theta$. The objective is to maximize the score function

$$J(\theta) = \mathbb{E}_{\pi_\theta}[\sum_t R_t] \quad (3)$$

where the gradient of the policy is defined by the Policy Gradient Theorem (Sutton et al., 2000) as:

$$\nabla_\theta J(\theta) = \mathbb{E}_{\pi_\theta}[\nabla_\theta \log \pi_\theta(a|s) Q_\pi(s, a)]. \quad (4)$$

This paper uses the actor-critic framework, where a network, namely the critic, learns the approximation of $Q_\pi(s, a)$ by temporal difference learning. To handle nonstationarity in the multiagent framework (Lowe et al., 2017), each agent's critic uses all agents' observations and actions for training.

Thus, the loss with respect to agent $i$'s policy parameterization is:

$$\nabla_{\theta_i} J = \mathbb{E}_{\pi_{\theta_i}}[\nabla_{\theta_i} \log \pi_{\theta_i}(a_i|o_i) Q_{\pi_i}(o_1, ..., o_N, a_1, ..., a_N)]. \quad (5)$$

## 4. Methods

This paper proposes three recurrent multiagent actor-critic models for partially observable and limited communication settings. The models only take in a single frame at each timestep. Because they cannot communicate all the time, they need a way to remember the last communication they received from their team, when they last transmitted a message and how their actions affect the communication budget over time. Recurrency acts as an explicit mechanism to do just that. Our models extend the multiagent actor-critic framework proposed by MADDPG to enable learning in a multiagent, partially observable, and limited communication domain.

## 4.1. Recurrent Multi Agent Actor

We perform the following updates using experience sampled $\sim U(\mathcal{D})$. An agent $i$'s replay buffer $\mathcal{D}$ contains tuples of experiences, where an experience at time $t$ contains: $(o_{i,t}, a_{i,t}, o'_{i,t+1}, r_{i,t}, h^p_{i,t}, h^p_{i,t+1})$. $o$ denotes agent $i$'s partial observations, $a$ its action resulting from $\pi_{i,t}(o_{i,t}, h_{i,t})$, $r_{i,t}$ the agent's reward, and $h^p$ the hidden state of the actor network before and after the selected action. Let each agent have a continuous policy $\mu = \mu_{\theta_i}$, and a target policy $\mu' = \mu'_{\theta'_i}$. $x \sim U(\mathcal{D})$ and $a \sim U(\mathcal{D})$ are placeholders for the state and action information of all agents from sampled experiences. Then, the policy's gradient is,

$$\nabla_{\theta_i} J(\mu) = \mathbb{E}_{\mathbb{U}(\mathcal{D})}[\nabla_{\theta_i} \mu(a_{i,t}|o_{i,t}, h^p_{i,t})$$
$$\cdot \nabla_{a_{i,t}} Q^\mu_i(x, a)|_{a_{i,t} = \mu_i(o_{i,t}, h^p_{i,t})}]. \quad (6)$$

The action-value function $Q^\mu_i$ is updated based on,

$$\mathcal{L}(\theta_i) = \mathbb{E}_{\mathbb{U}(\mathcal{D})}[((r_i + \gamma Q^{\mu'}_i(x', a'_j)|_{a'_j = \mu'_j(o_j, h^p_{j,t})}$$
$$- Q^\mu_i(x, a))^2]. \quad (7)$$

## 4.2. Recurrent Multi Agent Critic

We perform the following updates using experience sampled $\sim U(\mathcal{D})$. An agent $i$'s replay buffer $\mathcal{D}$ tuples of experiences, where an experience at time $t$ contains: $(o_{i,t}, a_{i,t}, o'_{i,t+1}, r_{i,t}, h^q_{i,t}, h^q_{i,t+1})$. We assume the same notation from before, where $h^q$ is the hidden state of the critic network before and after a selected action. The policy gradient is calculated as,

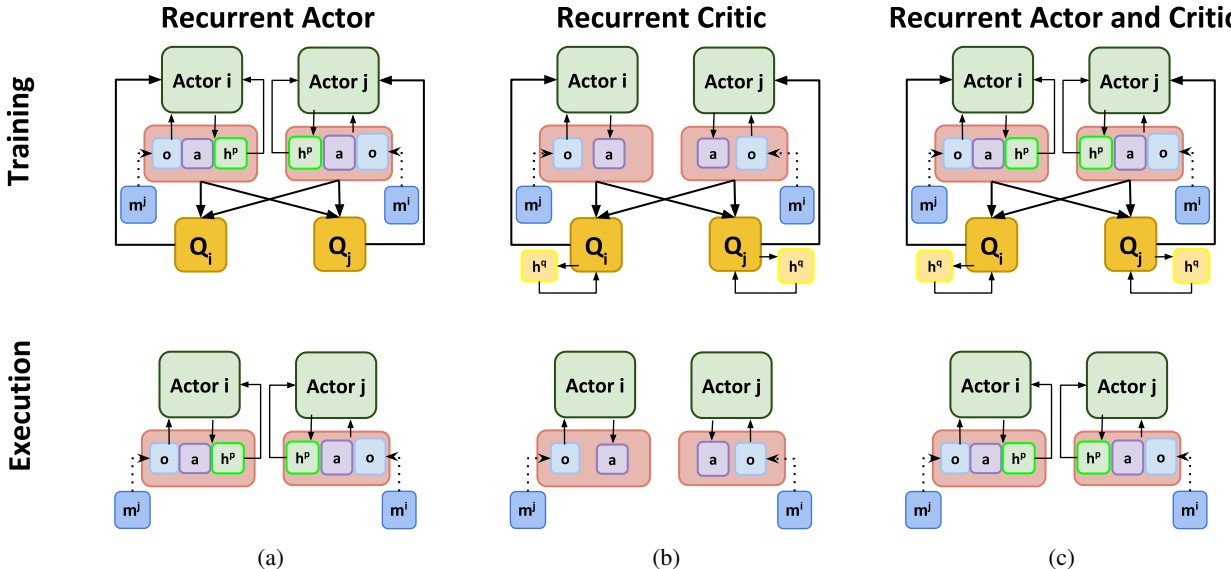

**Figure 1.** Illustration of the three recurrent models described in Section 4. 1a is the recurrent actor (actors maintain state $h^p$ over time), 1b is the recurrent critic (Q maintains $h^q$ over time), and 1c is the recurrent actor critic models used in the experiments. The top row shows the models during training, and the bottom row shows the models during execution. Actors communicate with each other and share information ($m$). If they decide not to communicate or have no communication budget left, an empty message is sent.

$$\nabla_{\theta_i} J(\mu) = \mathbb{E}_{\mathbb{U}(\mathcal{D})}[\nabla_{\theta_i} \mu(a_{i,t}|o_{i,t})$$
$$\cdot \nabla_{a_{i,t}} Q_i^\mu(x, a, h_t^q)|_{a_{i,t}=\mu_i(o_{i,t})}]. \quad (8)$$

The action-value function $Q_i^\mu$ is updated based on,

$$\mathcal{L}(\theta_i) = \mathbb{E}_{\mathbb{U}(\mathcal{D})}[((r_i + \gamma Q_i^{\mu'}(x', a'_j, h_{t+1}^q)|_{a'_j=\mu'_j(o_j)}$$
$$-Q_i^\mu(x, a, h_t^q)^2]. \quad (9)$$

### 4.3. Recurrent Multi Agent Actor and Critic

We perform the following updates using experience sampled $\sim U(\mathcal{D})$. An agent $i$'s replay buffer $\mathcal{D}$ tuples of experiences, where an experience at time $t$ contains: $(o_{i,t}, a_{i,t}, o'_{i,t+1}, r_{i,t}, h_{i,t}^q, h_{i,t+1}^q, h_{i,t}^p, h_{i,t+1}^p)$. We assume the same notation from before. The policy gradient is calculated as,

$$\nabla_{\theta_i} J(\mu) = \mathbb{E}_{\mathbb{U}(\mathcal{D})}[\nabla_{\theta_i} \mu(a_{i,t}|o_{i,t}, h_{i,t}^p)$$
$$\cdot \nabla_{a_{i,t}} Q_i^\mu(x, a, h_t^q)|_{a_{i,t}=\mu_i(o_{i,t}, h_{i,t}^p)}]. \quad (10)$$

The action-value function $Q_i^\mu$ is updated based on,

$$\mathcal{L}(\theta_i) = \mathbb{E}_{\mathbb{U}(\mathcal{D})}[((r_i + \gamma Q_i^{\mu'}(x', a'_j, h_{t+1}^q)|_{a'_j=\mu'_j(o_j, h_j^p)}$$
$$-Q_i^\mu(x, a, h_t^q)^2]. \quad (11)$$

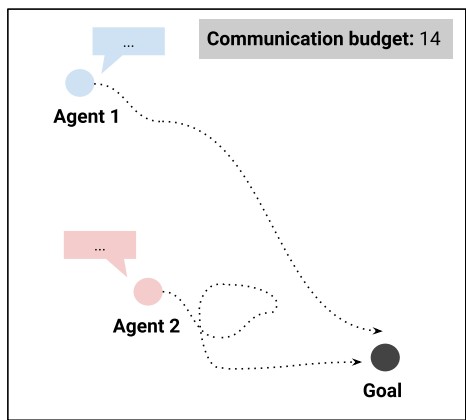

**Figure 2.** Simultaneous arrival task with $N = 2$ agents. The agents (blue, red) start at different distances from the goal (black), and their task is to arrive at the goal location simultaneously. A video can be found here.

## 5. Experiments

This section shows that the recurrent critic is critical for agents to learn a good policy from their partially observable states and under limited communication settings. The recurrent actor alone is not able to discover the right policy, however combined with the recurrent critic it reduces the variance in the reward performance. Our experiments use a *simultaneous arrival task*, where $N$ agents must arrive

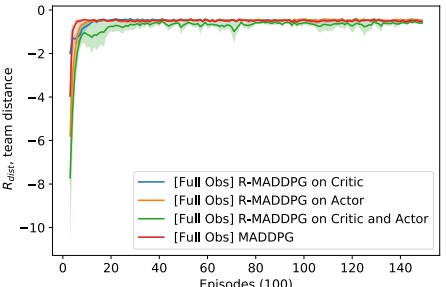
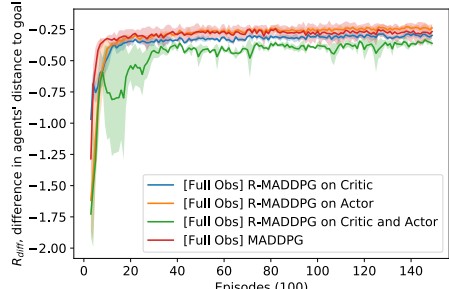

(a) Team *distance* reward, *fully* observable settings

(b) Team *difference* reward, *fully* observable settings.

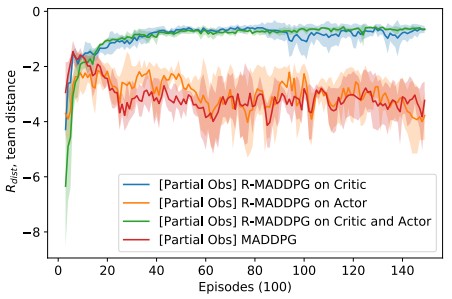
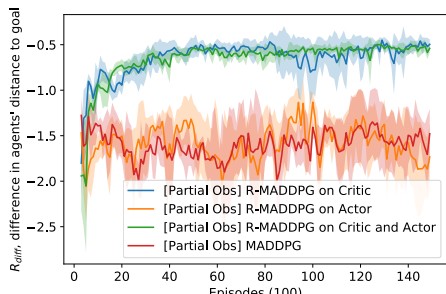

(c) Team *distance* reward, *partially* observable settings.

(d) Team *difference* reward, *partially* observable settings.

*Figure 3.* Reward performance in observability experiments. Under fully observable settings (top row), both MADDPG (red) and recurrent variants (green, blue, orange) perform similarly. Under partially observable (bottom row) settings, the recurrent actor (orange) and MADDPG (red) are unable to learn how to simultaneously arrive (d), and even how to move towards the goal (c). This demonstrates the importance of the recurrent critic in partially observable settings. For partial observability, the communication budget is set to 20 messages, shared between 2 agents over $\sim 100$ timesteps per episode.

at a goal location at the same time (extended from (Lowe et al., 2017), see example video here). In the fully observable environment, the agents know the positions of all the agents and the goal. In the partially observable environment, the agents only know their position and the goal; only if an agent decides to communicate, do other agents know its position. The partially observable domain is especially difficult for MADDPG because it is unable to keep a history of its previous partial observations; this renders it almost impossible for MADDPG to estimate the underlying system state.

This environment allows us to focus on the analysis of time-wise coordination among agents and multi-timestep communication use under different recurrent architectures. We investigate the effects of recurrency between MADDPG and R-MADDPG with the experiments below. For all the experiments we compare among regular MADDPG and these proposed networks from the Methods section. We both vary the observability (between full and partial observations) for the agents and vary the communication budget.

### 5.1. Experimental Setup

Let **s** denote an agent's fully observable state, containing this agent's position, $(p_x, p_y)$, the goal position $(g_x, g_y)$,

the communication message, $m$, is always the other agent's position, and a communication budget $c$. A partially observable state, $\mathbf{s}'$, contains the same state variables, however the communication message, $m'$ is either the other agent's position if the other agent communicated that timestep, or $(-1, -1)$ otherwise. That is,

$$\mathbf{s} = [p_x, p_y, g_x, g_y, m, c] \tag{12}$$
$$\mathbf{s}' = [p_x, p_y, g_x, g_y, m', c] \tag{13}$$

At each time step, each agent selects two types of discrete actions, one physical, $a^p \in \{\text{none}, \text{north}, \text{east}, \text{west}, \text{south}\}$, and one verbal, $a^v \in \{\text{communicate}, \text{silent}\}$.

The joint team reward function,

$$R = \underbrace{\sum_i d(p_i, g)}_{R_{dist}} + \underbrace{\sum_{pairs(i,j)} |d(p_i, g) - d(p_j, g)|}_{R_{diff}}, \tag{14}$$

encourages agents to individually reach the common goal position $g = (g_x, g_y)$ through $R_{dist}$, and encourages simultaneity through $R_{diff}$, where $d$ is Euclidean distance.

Throughout this section, we refer to $R_{dist}$ as the *team distance* and $R_{diff}$ as the *difference in agents' distances to goal*. These measurements are used in evaluating performance respectively on the left and right hand columns of

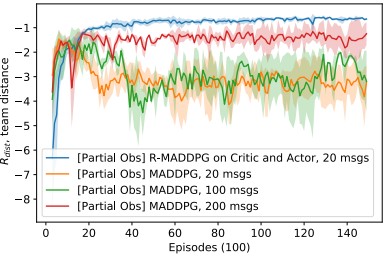 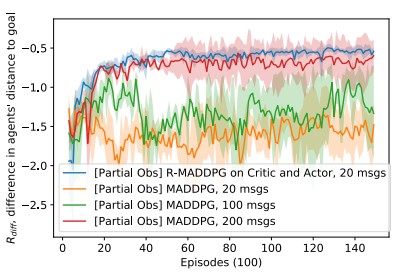

(a) Team distance to goal.  (b) Team difference to goal.

*Figure 4.* MADDPG's performance depends on the degree of observability. Decreasing the communication budget dramatically worsens MADDPG's performance in partially observable domain. These plots assume each episode is 100 timesteps. Thus, a shared communication budget of 200 messages means that both agents are able to communicate at every timestep during an episode. Yet, even with a 200 message budget that could enable full observability, MADDPG still performs worse than R-MADDPG which uses only 10% of the budget.

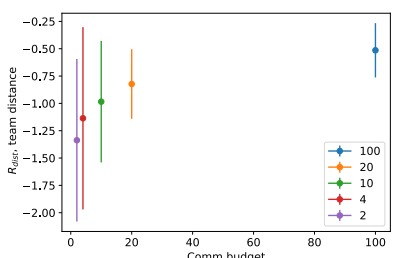 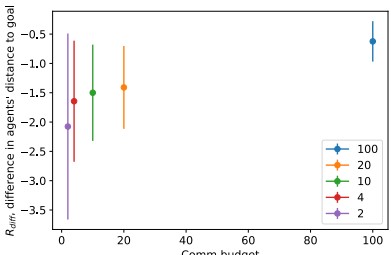

(a) Team distance to goal location over communication budget.  (b) Difference in agents' distances to goal over communication budget.

*Figure 5.* Execution performance over varying communication budget. The experiments assume R-MADDPG in a partially observable domain. The numbers indicate the amount of shared communication budget the agents had. We note that with decreasing budget the agents still learn how to move to the goal, and performance declines with decreasing budget, as expected.

Figure 3 and Figure 4.

The communication budget is shared between agents, and a full communication budget is consider to be $1.0$. If the communication budget is set to sending $x$ (total) messages, then the budget decreases by $\frac{1}{x}$ with every communication message. If no budget is given, i.e. no communication is allowed, the budget is set to $0.0$. No agent is allowed to communicate once the budget reaches $0.0$, and their messages are defaulted to a blank value $(-1, -1)$.

**Network architecture:** The networks contain three layers each with 64 units, where the first and last are fully connected layers and the middle layer is an LSTM layer. The first fully connected layer has an ReLU activation (Nair & Hinton, 2010).

**Hyperparameters:** The experiments assume an Adam Optimizer with a learning rate of 0.01, $\tau = 0.01$ for the target network updates, and $\gamma = 0.95$. The replay buffer size is $10^6$. We sample after every other 100 timesteps, and sample a batch size of 256 by episode. Training happens with 4 random seeds for all the experiments found above. All the hyperparameters will be set as the default in the open-sourced implementation of R-MADDPG.

## 5.2. Results

### 5.2.1. OBSERVABILITY

This section first explores whether the models are capable of learning in multiagent environments assuming complete observations, then learning in multi-agent environments assuming partial observations.

The experiments verify that both MADDPG and R-MADDPG variants perform equivalently well under fully observable settings in going to the goal (Figure 3a). R-MADDPG (in green, Figure 3b) does not converge as quickly in arriving simultaneously, and we hypothesize this is because it takes longer to learn if backpropagating through time in both in the actor and critic.

Under partially observable settings, the experiments illustrate the importance of the recurrent critic for learning a policy from partial observations and under a limited communication budget that, at minimum requirement, moves the agents towards the goal (Figure 3c, Figure 3d). Furthermore, the figures illustrate that the recurrent actor and critic learns more stably than only the recurrent actor model; we define stable learning by the reward mean fluctuations and

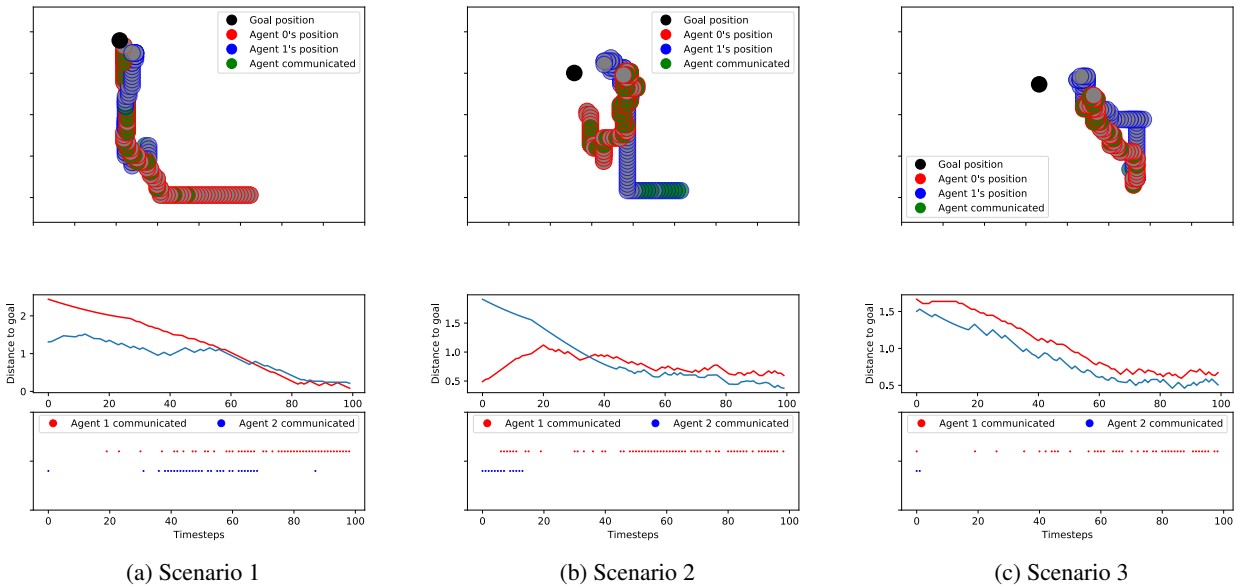

(a) Scenario 1         (b) Scenario 2         (c) Scenario 3

*Figure 6.* Example scenarios assuming R-MADDPG and shared communication budget of 50 messages (agents can only communicate $\sim 25\%$ of timesteps). Here is a video including these examples. Using communication, agents either hover around their location (blue agent in Figure 6a) or move away from the goal (red agent in Figure 6b) in order to synchronously arrive at the goal with the other agent. In cases where one agent dominates the communication (red agent in Figure 6c), the agents take longer to arrive to the goal even though they are initialized close to one another.

the reward variance, where these experiments assumed the same experiment stochasticity as described in section 5.1.

The experiments demonstrate that recurrent actor by itself performs similarly to MADDPG. It is unable to learn from a sequence of partial observations, not only how to simultaneously arrive at the goal (Figure 3d), but to even to go to the goal (Figure 3c). In other words, the recurrent actor provides insufficient information about the underlying task.

We hypothesize this is because the actor optimizes with respect to the critic (Eq. 6). A policy gradient taken with respect to $Q_i^{\mu}(x,a)|_{a=\mu_i(o_i,h_i^p)}$, a critic that cannot capture the partial observable dynamics of the environment, results in the actor converging to a poor policy. Thus, we believe that the RNN plays a more important role in the critic $Q_i^{\mu}(x,a,h_i^q)$ than in the actor $\mu_i(o_i,h_i^p)$.

### 5.2.2. COMMUNICATION BUDGET

This section investigates how well the models perform under different resource constraints by varying the communication budget shared by the agents. The communication budget dictates how many messages are allowed to be sent within a team of agents. We still assume the agents are in a partially observable environment, thus the agents must share information in order to arrive simultaneously at the goal. Video examples of R-MADDPG, which illustrate the communication use and physical movements of the agents, can be

found here.

The experiments verify that the poor performance seen in Figure 3c and Figure 3d by MADDPG is due to the insufficient communication budget which prevents MADDPG from having complete observations over the environment at every timestep. Figure 4 fixes the best performing model from Figure 3c and Figure 3d, namely R-MADDPG, and uses it as the best performing model. The graph increases the communication budget for MADDPG up to 200 messages, which means that every agent is allowed to communicate at each timestep of the episode. Only when this happens does the model's performance closely match R-MADDPG's performance under partially observable conditions.

An examination of how communication is used throughout an episode is in Figure 6, which identifies emergent coordination-communication behaviors that do not come across in the plotted aggregate statistics. Notably with limited communication, agents learn to either wait or move away from the goal in order to simultaneously arrive at the goal with the other agent (Figure 6a, Figure 6b). There exist edges cases, for instance Figure 6c, where agents are initialized closed to each other however one agent dominates the communication and the agents take longer to get to the goal.

The experiments also vary the communication budget on R-MADDPG to evaluate how sensitive the model is to de-

creased observability. The figures Figure 5a, Figure 5b display the reward performance (Eq. 14) over communication budget. As expected, the figures show that with increasing amounts of communication budget, the agents perform better at simultaneously arriving. Additionally, the performance variance decreases with increasing amount of communication budget in both plots. The plot illustrates the tradeoff between network bandwidth and team performance, which could be used to inform system-level design decisions in real-world applications.

## 6. Conclusions and Future Work

This paper proposes a recurrent multi-agent actor-critic model for coordination in partially observable, limited communication settings. This model is more applicable to real-world conditions since real-world settings are multiagent, partially observable and limited in communication. The experiments showed the recurrent critic is important for enabling R-MADDPG to handle partially observable environments. They also showed shown that R-MADDPG is capable of enabling coordination among agents in arriving simultaneously while varying the communication budget.

As future work, we hope to develop create more multi-agent coordination and communication scenarios and evaluate R-MADDPG in more other environments, such as the environments used in (Mordatch & Abbeel, 2017) and DeepMind's soccer environment from (Liu et al., 2019).

We also hope to expand it to coordination among heterogenous agents and explore the effects of the replay buffer parameters/settings in the multi-agent environments.

## 7. Acknowledgments

The authors would like to thank Dong-Ki Kim and Macheng Shen for their feedback on the paper's draft and interesting discussions. This work was supported by Lockheed Martin.

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
