# OpenReview forum: "R-MADDPG for Partially Observable Environments and Limited Communication"
_ICML.cc/2019/Workshop/RL4RealLife — RL4RealLife 2019_

### Official Review · AnonReviewer1 · 2019-05-22
**Interesting work enough to be presented at the workshop but more discussion on the results is required**

**Rating:** 5
**Confidence:** 5

**Review:**

This paper extends MADDPG (Lowe et al., 2017) by including partial observability and recurrent neural networks. The authors include RNN in 1) Actor only, 2) Critic only, and 3) both Actor and Critic and the performance is observed as 3>2>>1. Based on such experimental results, the authors claim that the RNN plays a significant role when it is employed in Critic.

The fundamental motivation for the real-world application is well described in Introduction, e.g., neither full observation nor unlimited communication is not always allowed. This paper in general well-written and the reviewer enjoyed the video on the paper webpage. However, the reviewer has a few minor comments that can be helpful for a future revision of this paper.

1.	More reasoning/ explanation about why the RNN in critic is more important than actor. This seems very clear in the experimental results, but there is no enough about of discussion about why. Since this result is very interesting but not straightforward, the authors should put more efforts on elaborating it.
2.	In Figure 4, there is little performance gap between 20 messages and 100 messages, but the gap becomes large between 100 messages and 200 messages. This is a counter-intuitive result because the impact of a unit resource is usually higher when it is scarce compared to when it is abundant. Thus, it is more natural to expect that the higher gap between 20 and 100 than 100 to 200. There are additional results regarding the number of messages in Figure 6, but it is hard to tell with the current message set (i.e., 0, 2, 4, 20, 100). It would be interesting to see the execution performance (y-axis) over the number of messages (x-axis).
3.	In Figure 4, R-MADDDPG on Critic and Actor with 20 messages (blue) outperforms full observation MADDPG (i.e., red - Partial observation in MADDPG with 200 messages). This is a fascinating result, but more elaboration is required to explain it.

---

### Official Review · AnonReviewer2 · 2019-05-24
**A recurrent extension of MADDPG**

**Rating:** 2
**Confidence:** 4

**Review:**

- quality, clarity, originality and significance:
The paper is well organized. The authors' contribution is using recurrent networks to provide "memory" and compress communication messages in a MARL setting. The idea is original but not completely novel (see my argument #2 in "cons"). The emphasis on "real life" application is not specific.

- pros:
The paper is well written. The related work and background sections are fairly comprehensive and very clear. The illustration in Figure 1 and equations in methods section did a great job on describing their proposed R-MADDPG framework. The experiment was conducted in a careful way, where they compared different performances between MADDPG and R-MADDPG with several settings.

- cons:
1. the authors mentioned that we are facing partial observability, nonstationarity, and limited communication between agents in real world RL applications. but they did not provide specific application examples or specific motivating examples. Also, the experiment "simultaneous arrival" with 2 agents is not a sufficiently "real world" application. For example, some real world applications do have full observability (or we can safely assume that), while some real world applications do not have communication at all.
2. The idea of using RNN in MARL is not completely novel in MARL literature. The authors may want to consider a more comprehensive literature survey to better identify their unique contribution. The idea of using RNN for message passing or communicating MARL is novel. For example, other than Foerster's 2016 paper (message passing, or the authors can consider to cite his work "Learning to Communicate with Deep Multi-Agent Reinforcement Learning"), his 2018 work on "Counterfactual Multi-Agent Policy Gradients" used GRU to compress histories (though their work is not using message passing). Similarly, the authors' MIT group had a work in 2018 of using LSTM in a non-communicating MARL: "Motion Planning Among Dynamic, Decision-Making Agents with Deep Reinforcement Learning".
3. The colors and density of these dots in Figure 5 illustration are difficult to read
4. The figures are not in vector graphics, e.g. the resolution in Figure 3 and Figure 4 are not clear.
5. The page limit is not wisely used. Given 8 pages, the authors may want to consider to put some content as Appendix.

---

### Decision · Program_Chairs · 2019-05-28

Accept